

# Contrasting organic aerosol molecular composition between the urban and agricultural environment of the Po Valley

Luca D'Angelo[1,2], Florian Ungeheuer[1], Jialiang Ma[1,3], Luca Ferrero[4], Cristina Colombi[2], Eleonora Cuccia[2], Umberto Dal Santo[2], Beatrice Biffi[2], and Alexander L. Vogel[1]

[1]Institute for Atmospheric and Environmental Sciences, Goethe University Frankfurt, Frankfurt am Main, Germany
[2]Environmental Protection Agency of Lombardy Region (ARPA Lombardia), Milan, Italy
[3]Department of Chemistry, Aarhus University, Aarhus, Denmark
[4]GEMMA and POLARIS research Center, Department of Earth and Environmental Sciences, University of Milano-Bicocca, Milano, Italy

**Correspondence:** Luca D'Angelo (l.dangelo@arpalombardia.it), Alexander L. Vogel (vogel@iau.uni-frankfurt.de)

**Abstract.** The molecular composition of organic aerosol in the Po Valley remains largely unexplored, despite contributing approximately half of the fine aerosol mass. Molecularly-resolved analysis of the organic fraction is essential for understanding the sources and formation processes of organic aerosol in detail. Here, we investigated one year of $PM_{2.5}$ filter samples from a rural agricultural and an urban environment. We used liquid chromatography and high-resolution mass spectrometry with subsequent non-target analysis of 250 samples. Time-series analysis allowed for the grouping of detected organic compounds using a clustering algorithm, enabling a tentative source attribution. The most populated clusters consist mainly of CHO- and CHOS-containing compounds, attributed to oxidation products of biogenic emissions. They account 31 % and 26 % of the total intensity in the urban and agricultural sites, respectively, and peak during summer. Combustion-related clusters, enriched in nitrogen-containing compounds, contribute approximately 35 % of the total intensity at both sites. A fraction of these molecules are highly conjugated compounds that appear during winter as primary contributors to light-absorbing organic aerosol. Furthermore, we identified site-specific compound clusters, either at the urban or at the agricultural site. At the latter, we found pesticides strongly influence the overall molecular composition, peaking in May when $PM_{2.5}$ concentration is at its lowest level. This suggests potential toxicological effects despite apparent good air quality conditions. Our results represent the first molecular-level characterization of organic aerosol in the Po Valley, emphasizing the need to assess its composition for a better understanding of its environmental and health impacts.

## 1 Introduction

The chemical characterization of atmospheric aerosols is essential for developing effective mitigation strategies targeting the most critical sources of air pollution. It is well-established that aerosols affect human health, particularly among the most vulnerable groups of the population (Fann et al., 2018; Kioumourtzoglou et al., 2015; Mikati et al., 2018; Bell and Ebisu, 2012). High exposure of aerosols causes respiratory apparatus cancer (Chang et al., 2022), increases the risk for cardiovascular diseases (Wu et al., 2020; Liang et al., 2022; Chen et al., 2022; Peralta et al., 2022; Xie et al., 2018; He et al., 2022a), and





decreases global cognitive performance (Weuve, 2012; Saenz et al., 2018; Ailshire et al., 2017; Harris et al., 2015) resulting in increased hospitalizations burdened on national health systems (Lee et al., 2019).

Furthermore, aerosols interact with radiation, leading to a modification of Earth's energy balance. Ammonium nitrate and am-
monium sulfate aerosols increase light scattering, which causes atmospheric cooling (Zhu et al., 2015; Adams et al., 2001). Drugé et al. (2019) estimated that ammonium nitrate aerosols cause a mean direct radiative effect at the top of the atmosphere of $-1.4$ W m$^{-2}$ over Europe, with a local maxima of $-5$ W m$^{-2}$ over the Po Valley, Italy, in all sky conditions. In contrast, black carbon generated by incomplete combustion is a strong light absorber and has a positive radiative effects ($+0.0$-$4.0$ W m$^{-2}$ over Europe (Nordmann et al., 2014)), locally increasing air temperature (Ramanathan and Carmichael, 2008). In addition, other
light-absorbing aerosols (LAAs) are organic aerosols (OA) formed by compounds rich in chromophores (Zeng et al., 2020), which strongly absorb radiation in the ultraviolet and visible region. These compounds are collectively referred to as "brown carbon" (BrC) due to the color that aerosol samples rich of this fraction assume. Ferrero et al. (2018) estimated BrC heating rate in the Po Valley accounts for $12.50 \pm 0.06$ % of the total, leading to a heating of $0.20 \pm 0.01$ K d$^{-1}$ in cirrostratus cloud conditions and $0.020 \pm 10^{-3}$ K d$^{-1}$ under conditions of complete cloud cover (8 oktas) (Ferrero et al., 2021).

Molecular-level characterization of OA is becoming increasingly important to infer aerosol physicochemical properties, toxico-logical effects, and their sources (e.g. biogenic vs. anthropogenic, primary vs. secondary OA). This is undoubtedly challenging due to the number of compounds involved. Recently, Thoma et al. (2025) have observed that even at a rural site, OA system is composed of several thousands of compounds affected by seasonality and short- and long-transport events, with biogenic secondary OA (BSOA) representing about 70 % of compounds and 30 % attributed to anthropogenic SOA (ASOA). The source
attribution is even more important for air pollution hot-spot regions. Because of this, regions like the North China Plain (Zhang et al., 2023b; Zhu et al., 2021; Li et al., 2019, e.g.) and the Indo Gangetic Plain (Panda et al., 2025; Gupta et al., 2022, e.g.) have been extensively studied. For example, Kumar et al. (2022) reported that about 60 % of OA mass has a primary origin in New Delhi, due to hydrocarbon-like OA (HOA) and primary biomass burning OA (BBOA). SOA were instead attributed to aromatics, BSOA, aged BBOA and mixed-urban SOA. Kapoor et al. (2024) pointed out the importance of LAAs, estimat-
ing that daily average BrC absorption contributes between 18 % and 42 % to near-ultraviolet absorption over India. Another well-studied pollution hot-spot is the Po Valley, in the Northern part of Italy. Colombi et al. (2024) have shown that aerosol ho-mogeneously affects the basin with very similar values in both concentration and chemical composition. The authors analyzed eleven years of particulate matter (PM) samples collected in an urban (Milan, the most populated city in the Po Valley) and rural (Schivenoglia, in a low-intensity agricultural region, about 150 km south-east of Milan) site, finding that both PM and
secondary inorganic aerosols constituents show very similar concentrations along the year. Many studies have been published about the source apportionment in the Po Valley. For example, Perrone et al. (2012) performed a source apportionment on PM$_{2.5}$ samples collected at urban and rural sites of the Po Valley over a three-year period (2006-2009), highlighting the strong contribution of traffic and biomass burning (BB) at both sites, and a residual contribution of secondary organic carbon (OC). Bernardoni et al. (2013) used $^{14}$C to distinguish OC and elemental carbon (EC) from fossil to modern sources on PM$_{10}$ samples
collected in Milan during wintertime. According to their findings, OC$_{fossil}$ represent about 30.8 % of total carbon, whereas primary wood burning OC 14.8 %. In addition, the authors estimated that about 37.5 % of the total carbon is due to additional



modern sources. Later, Daellenbach et al. (2023) estimated that primary BBOA accounts for 37 % during wintertime, with a lower contribution during summertime, which is instead dominated by SOA, predominantly low-oxygenated compounds from both biogenic and anthropogenic sources. Decesari et al. (2014) performed an intensive campaign during summer 2009 in San

Pietro Capofiume, a rural background site in the south-eastern Po Valley area. Based on their observations, HOA contributed, on average, to 38 % of the OA mass, while cooking OA and semi-volatile oxygenated OA accounted for 21 %, primarily due to aging processes occurring within the Po Basin. A substantial fraction (41 %) of OA was instead attributed to transalpine transport. Paglione et al. (2020) estimated that at the aforementioned rural site, SOA account for 83±16 %. In contrast, at an urban site approximately 35 km far, this percentage decreased to 60±16 % due to the higher contribution of local emission of

HOA. At the same site, Costabile et al. (2017) observed that BrC associated with SOA formation peaks in the droplet mode (400-700 nm) during fall and winter. The authors attributed their formation to aqueous-phase processes, in agreement with previous findings regarding water-soluble SOA from BB sources (Gilardoni et al., 2016). At other rural and urban sites within the Po Valley, approximately 220 km northwest of the aforementioned location, Gilardoni et al. (2020) observed that more than 50 % of the BrC is water-insoluble, likely composed of aromatic compounds associated with primary aerosols from wood

combustion.

Promising information to infer source attribution is provided by OA molecular characterization by ultra-high-resolution mass spectrometry (UHRMS) after soft ionization methods. Due to the high mass resolution and accuracy it is possible to derive molecular formulae from the exact mass. In addition, the chromatographic separation at the front of the mass spectrometer helps to resolve isomers. In this regard, many authors have already attempted to describe an aerosol system coupling ultra-high

performance liquid chromatography (UHPLC) to soft-ionization UHRMS. For example, Mazzi et al. (2024) recently used a high-performance anion-exchange chromatograph coupled with a triple quadrupole mass spectrometer with an electrospray ion (ESI) source to detect and quantify plant protection agents in Mestre, near Venice, on the eastern side of the Po basin. Among the trace compounds found, the authors estimated glyphosate concentration of 0.05 ng cm$^{-3}$ in PM$_{10}$. Many authors have begun to take advantage of the high resolution and accuracy of mass spectrometers, such as Orbitrap, to explore the complexity

of OA without targeting specific molecules (e.g., Hildmann and Hoffmann, 2024; Song et al., 2023; Divisekara et al., 2023). Non-targeted analysis allows for inferring the overall properties of OA and has the potential to identify relationships between groups of compounds and their sources or other endpoints. To our knowledge, no work applied these techniques to OA in the Po Valley.

In this work, we applied the aforementioned techniques to two sets of fine atmospheric particulate matter samples (PM$_{2.5}$)

collected in Milan (MI) and Schivenoglia (SKI), both within the Po Valley, during 2021. Although organic carbon (OC) concentrations are frequently comparable between the agricultural (SKI) and urban (MI) monitoring sites (90° percentile of absolute difference: 2.0 $\mu$g m$^{-3}$), the molecular characterization of OA reveals marked qualitative differences in molecular aerosol composition, resulting in different light-absorbance spectral behavior too. Furthermore, the implementation of a compound-level clustering analysis allowed for the source attribution of potentially light-absorbing BrC constituents, elucidating distinct

source profiles between the two environments.



## 2 Method

### 2.1 Sampling

PM$_{2.5}$ was collected at two different locations, Milan-Pascal (45° 28' 42.0348 N, 9° 13' 54.03 E) and Schivenoglia (45° 1' 0.336 N, 11 ° 4' 34.0032 E), both belonging to the ARPA Lombardia (the regional environmental protection authority) air

quality network. The first one (hereinafter "MI") is an urban background site, located in a small park next to a low-traffic street of the University of Milan campus. Milan is the most populated city in the Northern part of Italy and is located in the center of the Po Valley. The second site (hereinafter "SKI") is a rural background site about 150 km far from MI, surrounded by agricultural fields and few animal husbandries.

At both sites, low volume samplers (SKYpost PM, TCR-Tecora, or Lifetek PMS, Megasystem, 16.67 Lpm) were used to

collect the PM$_{2.5}$ samples on quartz fiber filters (Pall TissueQuartz, Ø=47 mm) for 24 hours since 2014. On a weekly basis, the filters were collected and stored in the darkness at low temperature (about 4 °C). For this work, we chose to investigate samples collected in 2021. Therefore, a total of 132 samples for MI site and 117 samples for SKI site were used for the chemical and molecular characterization. The covered period ranges from the end of January 2021 to the end of the same year. Moreover, additional data, such as meteorological parameters (air temperature, global radiation, relative humidity) and gaseous

concentration of NO$_x$ (NO+NO$_2$), benzene and toluene, were provided by ARPA Lombardia.

### 2.2 Chemical analysis

Water-soluble inorganic compounds and levoglucosan were quantified by ion chromatography (Metrohm 930 and 881) by extracting 1.5 cm$^2$ quartz filter punches in ultrapure water for 20 min in an ultrasonic bath. A second punch was used to quantify EC and OC through thermo-optical analysis (Sunset Laboratory Inc., Tigard, OR, USA) based on the EUSAAR2 protocol.

For the organic phase investigation, a 12 mm diameter round punch was used for OA extraction in 180 $\mu$L of a 10 % acetonitrile (ACN) solution (Merck KGaA and Optima LC/MS Grade, Thermo Fisher Scientific Inc. and Milli-Q water Reference A+). The extraction was carried out using an orbital shaker (300 rpm) for 20 minutes. To the extracted and filtered (non-sterile PTFE Syringe Filter, Thermo Fisher Scientific Inc.) solution of 80 $\mu$L , an internal standard (10 % of the final extracted volume) consisting of isotopically labeled benzoic acid (C$_6$H$_5^{13}$CO$_2$H, 99 atom % $^{13}$C, SigmaAldrich) and caffeine ($^{13}$C$_3$C$_5$H$_{10}$N$_4$O$_2$, 99

atom % $^{13}$C SigmaAldrich) was added.

The analytes were separated by injecting 10 $\mu$L of each sample into a UHPLC with a reverse phase column (CORTECS T3, 120 Å, 2.7 $\mu$m, 3 mm × 150 mm, Waters, C-18) system. The mobile phase, ultrapure water and ACN, was maintained at a constant flow rate of 0.4 mL min$^{-1}$ through the C-18 column heated at a constant temperature (40 °C). The gradient program was set as follows: 0-1 min at 1 % (v/v) ACN, 1-16 minutes with a linear increase to 99 % ACN, 16-16.5 min at a constant 99 % ACN,

16.5-17.5 min with a linear decrease back to 1 % ACN, and 17.5-20 min at 1 % ACN to re-equilibrate the column. A heated electrospray ion source (HESI-II Probe, Thermo Fisher Scientific Inc., operating in both positive and negative modes) was used to ionize the analytes before entering the mass spectrometer (high-resolution hybrid quadrupole-Orbitrap mass spectrometer, Q-Exactive Focus, Thermo Fisher Scientific Inc.). Ions in 75-750 *m/z* range were detected in both positive and negative ioniza-



tion mode in full-scan MS with a resolution of 70,000 at *m/z* 200. Data-dependent tandem mass spectrometry (ddMS$^2$) with a
resolution of 17,500 was used at higher-energy collision energies (15, 41/30 (negative/positive mode), 50 eV). As quality as-
surance, the sample analysis was periodically supplemented with mixed standards to evaluate the performance of the analytical
system. A previous evaluation was also performed in Thoma et al. (2022).

## 2.3 Light-absorbance measurements

A subset of sample extracts was used to assess the light-absorbance to compare samples, which were collected at the two sites
on the same days and exhibit very similar OC concentrations and contribution to PM$_{2.5}$ (within a tolerance of 2 $\mu$g m$^{-3}$ and 5
%, respectively). For this purpose, 10 $\mu$L of sample extracts were directly injected without chromatographic separation into a
diode array detector (DAD, Thermo Scientific, Vanquish, VH-D10-A) with a liquid carrier (99 % ACN, 300 $\mu$L min$^{-1}$). The
light absorbance was measured with 20 Hz collection rate over the wavelength range of 230-650 nm. Solvent blank with the
same amount of the internal standard injected in the sample extracts was used to remove the absorbance background. Finally,
to account for the absence of a quantification of the OC mass extracted, we calculated the Absorbance Angstrom Exponent
(AAE) for three wavelength ranges, that are 230-250 nm, 250-400 nm, and 400-650 nm. To this end, a linear interpolation was
performed between the logarithm of absorbance and that of wavelength. The absolute value of the resulting slope represents
the light-spectral behavior of the extracts.

## 2.4 Data analysis

### 2.4.1 OA non-target analysis

The chromatograms and acquired spectra were processed using Compound Discoverer software (Thermo Fisher, version 3.3.2).
We considered spectra with retention time in the range 0-16.5 min and with a total intensity threshold of $10^4/10^5$ (negative/pos-
itive mode). Retention time alignment was performed using an adaptive curve with a maximum absolute shift of 0.2 minutes.
Each peak signal was extracted with a mass tolerance of $\pm 2$ ppm only if it appeared at least for 5 scans. Base ions for compound
detection was set to [M-H]$^-$ and [M+H]$^+$ for negative and positive mode, respectively. Molecular formulae were attributed
constraining the number of elements (min: C#1, H#1; max: C#90 H#190 Br#3 Cl#4 N#4 O#20 P#1 S#3), the DBE range (0-
40), and the hydrogen-to-carbon ratio range (H/C, 0.1-3.5). The acquired MS$^2$ spectra were compared with the Advanced Mass
Spectral Database mzCloud (HighChem LLC, Slovakia) and the *Aerosolomic* database, described in Thoma et al. (2022). A
match with these libraries allowed to increase the identification confidence from Level 4 (L4, unequivocal molecular formula)
to L2 (probable structure), based on the classification described in Schymanski et al. (2014). Detailed workflows are given in
Tables S1 and S2 for negative and positive mode, respectively.



### 2.4.2 OA molecular characterization and hierarchical cluster analysis

After blank correction, the dataset was reduced considering only the compounds with a signal intensity (SI) higher than $10^4$ and $10^5$ in negative and positive mode, respectively, and with a signal-to-blank and a peak rate higher than 5. Moreover, to analyze the chromatographically resolved features only, we considered the retention time range between 1.55 min (void time: 1.44 min) and 16.50 min. The resulting list of molecular formulae were compared with the PubChem database (Kim et al. (2024), last accessed June 2024) to remove only potentially non-existent or impossible formulae. Assigned molecular formulae were used to group the features into families based on the elemental composition: CHO, CHNO, CHOS, CHNOS, CHN, and CHOP. Features without an assigned formula or not falling into the aforementioned categories were grouped under the category 'Others'. To compare SI among the samples, each individual signal was scaled by the air volume sampled through the filter. Finally, to account for the different instrumental response in terms of signal peak intensity due to the ionization modes, the SI were normalized (nSI) to the maximum value found for each polarity (see Eq. 1 in Supplement).

The non-target screening approach requires the identification of descriptors to infer the properties or characteristics of detected molecules, which can aid in their characterization. For each formula, we calculated the average carbon oxidation state ($OS_c$) with the simplified method described in Kroll et al. (2011) (Eq.1), the double bond equivalent value (DBE, Eq.2), that is the number of rings and double bonds in a molecule, the aromaticity index ($X_c$), as proposed in Yassine et al. (2014) (Eq.3,). In this work, to infer temporal averages, we used the peak abundance-weighted average parameter approach, as already shown in Wang et al. (2021).

$$OS_c = 2 \times \#O/\#C - \#H/\#C \tag{1}$$

$$DBE = \#C - \#H/2 + \#N/2 + 1 \tag{2}$$

$$X_c = \frac{3 \times [DBE - (p \times \#O + q \times \#S)] - 2}{DBE - (p \times \#O + q \times \#S)} \tag{3}$$

In Eq.3, $p$ and $q$ represents the fraction of oxygen and sulfur atoms involved in the $\pi$-bond structures. Since their values depend on the category of compounds, we used the same approach proposed in Kourtchev et al. (2016),Wang et al. (2017), and Tong et al. (2021), where $p=q=0.5$ for compounds detected in HESI(–), while $p=q=1$ for formulae detected in HESI(+). In addition, the value of ($p\times\#O + q\times\#S$) is rounded down to the lower integer whether #O or #S are odd for compounds detected in negative mode according to Yassine et al. (2014).

It is worth stating here as well that the analytical approach used in this work is not suitable for the quantification of each compound. Although the SI is proportional to the concentration in the analyzed sample, soft ionization techniques have different





ionization efficiencies for each analyte. Theoretically, it would be possible to measure the ionization efficiency of each compound; however, the enormous number of molecules contained in an atmospheric aerosol sample makes this process practically impossible. Moreover, many compounds present in the atmosphere are completely unknown and, therefore, no analytical standard is available. This issue led some authors (e.g., Ma et al., 2022; Divisekara et al., 2023) to use surrogate compounds to represent a broader group of molecules and to assume the same concentration-peak area intensity response for these as for the

measured compound. Although, to our knowledge, this is the best approach currently available, we believe that the number of surrogates required would be too large for the purposes of this work. For this reason, we have limited our discussion to considering peak area intensities as a proxy for atmospheric concentration, i.e. to compare their relative importance in the samples. A hierarchical clustering algorithm was chosen to reduce the complexity of the system (Qi et al., 2019; Priestley et al., 2021; Thoma et al., 2022). We used the software MATLAB (The MathWorks, vers. 2024a) independently on the two datasets after

removing all the features with an occurrence lower than 10 %. We standardized (through centering) each feature along its time-series and used the Euclidean distance as a metric to calculate the similarity among the compounds and the Ward linkage method to create the tree. Finally, the number of clusters for each site was chosen based on the similarity metric, the resulting fingerprint of each cluster, its interpretability using available markers in the literature, and its correlation with external data, such as EC, soluble potassium, levoglucosan, and $NO_2$.


## 3   Results

### 3.1   Seasonality of organic aerosol molecular composition

The calculation of the normalized total signal intensity (nTSI, Eq. S2), i.e. the sum of the nSI of individual compounds in each sample, allowed us to compare the seasonal variability at each site. Both time series sites exhibit peak values during winter

and summer (Figure 1). A similar trend is also visible in the OC concentration quantified in the $PM_{2.5}$ samples from both sites, although the increase during the summer period, mainly due to aliphatic CHO and CHOS compounds, is not fully reflected in the OC values. This leads to a poor correlation ($R^2$=0.280 and 0.466 for MI and SKI, respectively) between nTSI and OC on an individual samples basis. Highly oxygenated compounds dominate during summer, with an average $OS_c$ of 0.91±0.06 for MI and 0.89±0.04 for SKI in July. Figure 1 indicates that nTSI has a stronger contribution in aromatic compounds, especially

belonging to CHNO and CHO families, during winter. In this regards, both DBE and $X_c$ index show minimum values during summer (2.54±0.14 and 0.37±0.09, respectively in July) and maxima in winter (3.40±0.34 and 0.87±0.15, in February). While no clear temporal trend is observed for P-containing compounds in Milan, two periods of notable contribution from this family to the nTSI are identified at the agricultural site. The annual pattern shows that this compounds contribution increases from March to early May and from late September to the beginning of December, peaking in samples collected in April and

November.

    Retrieving light-absorption information via a molecular-derived proxy, we use the DBE/#C ratio to identify compounds of light-absorbing aerosols (LAAs), setting limits for this range between 0.5 (for polyenes) and 0.9 (for fullerene-like hydro-




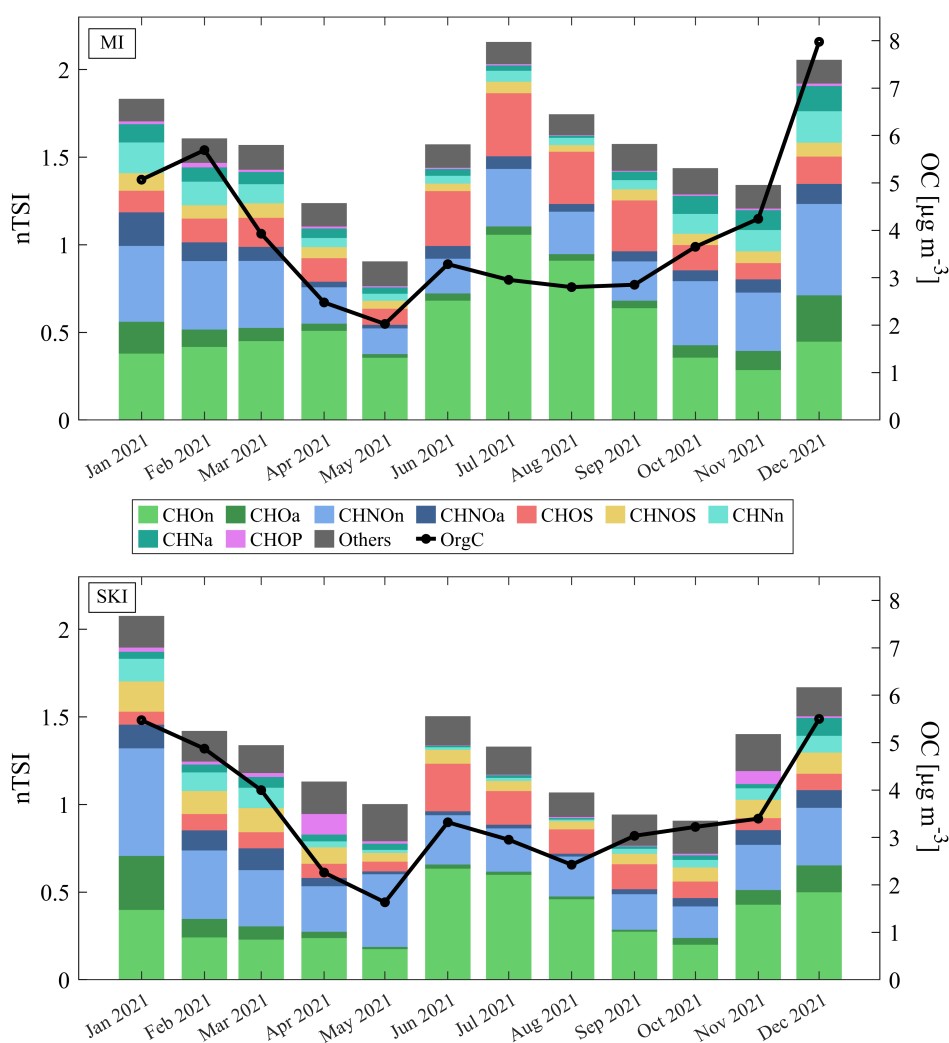

**Figure 1.** Time series of the monthly average nTSI for the groups of compounds detected in the MI (top plot) and SKI (bottom plot) samples. Darker colors represent the nTSI of aromatic compounds ("-a"), distinguishing them from the lighter colors for aliphatic ones ("-n") based on their $X_c$ values.





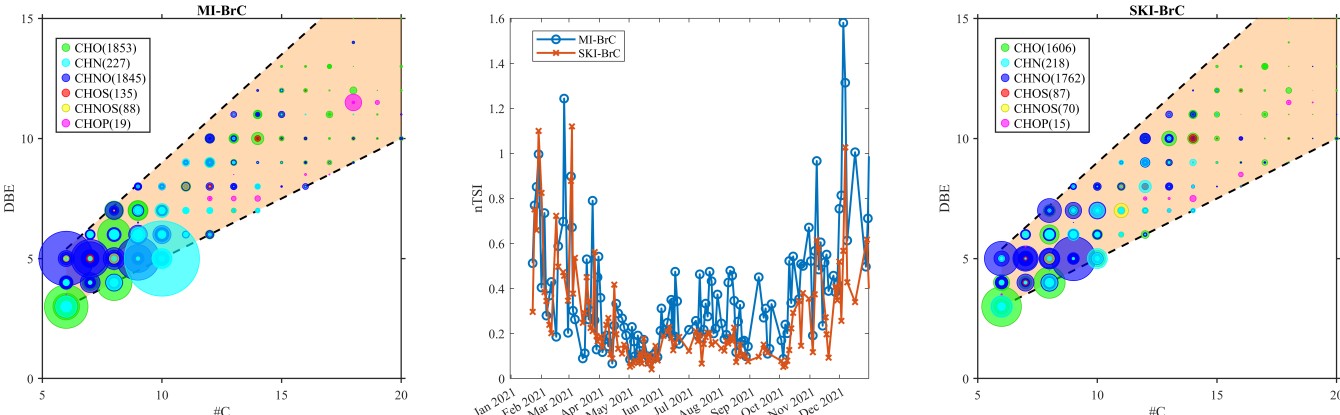

**Figure 2.** Fingerprints of potential BrC compounds at MI (left) and SKI (right) sites and their nTSI time series over the year 2021. The orange-shaded area delineates the region in which potential BrC compounds are located.

carbons) (e.g., Lin et al., 2018; Laskin et al., 2015). Using these criteria, we selected molecules from our dataset that likely contribute to BrC (Figure 2). Organic LAAs show maxima nTSI values during winter (0.73±0.36 at MI and 0.53±0.24 at

SKI). This is in agreement with Bluvshtein et al. (2017) and Budisulistiorini et al. (2017) describing that emission from wildfires, peat fires, agricultural residue burning, and residential combustion substantially contribute to BrC. While some of these sources may remain active during summertime, photobleaching and ventilation of the Po Valley are more efficient during summertime (Chen et al., 2021). Consistently, we observe lower values of the BrC-related molecules in summer (0.27±0.11 at MI and 0.16±0.04 at SKI). The higher values in nTSI observed at the urban site in the warm season indicate either more active

emission sources or air-quality conditions that more efficiently promote chromophores formation. Indeed, the plots in Figure 2 show that the urban site (left) exhibits a larger number of BrC-related compounds (+409) compared to the rural site (right). From a qualitative perspective, the seasonal BrC plots for the two sites (Figure S1) confirm a different composition of LAAs. When averaging intensity by compound families (Figure S2), it is observed that BrC at the urban site contributes more strongly to nTSI, mainly due to CHN species (8.8 % vs 3.7 % at MI and SKI, respectively, in summer; 17.1 % vs 7.9 % in winter),

whereas the agricultural site is more influenced by CHNO (23.9 % vs 33.9 % and 46.2 % vs 51.5 % in summer and winter, respectively). Similarly to the temporal pattern shown in Figure 1, at both sites the contribution of wintertime BrC is mainly due to N-containing compounds. At the same time, the nSI-weighted average shows that in the coldest months BrC has the lowest O/C ratio, suggesting a low oxidation regime affecting BrC molecular composition (Figure S3). On the contrary, the lower summertime peak in BrC nTSI is mainly ascribable to highly oxidized compounds (O/C>0.5) with a probable secondary

origin (Gilardoni et al., 2020).

In addition to the seasonality of elemental-composition groups and the BrC-related molecules, we focus on few individual tracers to confirm our hypothesis concerning their seasonal patterns. For instance, the later-generation product of monoterpene oxidation, that is 3-methyl-1,2,3-butane-tricarboxylic acid (MBTCA), peaks in summer. This is consistent with the higher monoterpene emission and stronger oxidative conditions from May to September. On the opposite, the CHNO compound ni-



trocatechol, a well-known combustion product from BB (Fredrickson et al., 2022), shows the opposite trend. The time series of these compounds, whose identification was allowed by the comparison with analytical standards (identification level 1, L1, based on Schymanski et al. (2014)), are shown in the Supplement (Figure S4).

Based on the blank-corrected annual average SI, we ranked the compounds with the highest values for the most representative molecular families (that are CHO and CHNO), which alone represent 60 % of the nTSI. Within the CHO(–) family, the

formulae $C_4H_6O_5$, $C_8H_{12}O_6$, and $C_6H_8O_7$ exhibit the highest average intensities. The first two compounds show a distinct seasonal trend, with intensities peaking in spring and reaching a maximum during the summer, followed by a decline during autumn. This pattern suggests that the conditions favorable for the emission or formation of these compounds are most prominent in summer. Together with the aforementioned MBTCA (L1, $C_8H_{12}O_6$), the compound with the formula $C_4H_6O_5$ and its pronounced increase during summer suggest a likely monoterpene-derived biogenic origin (L4, as suggested by Thoma

et al. (2022) and Chen et al. (2020) and the literature within). In contrast, the third-highest intensity compound, $C_6H_8O_7$, appears to be an anthropogenic SOA tracer (ASOA) derived from the oxidation of benzene (L4, Cheng et al. (2021)). For the CHNO(–) molecular family the highest signals occur during wintertime. At both sites, nitrocatechol (L1, $C_6H_5NO_4$) ranks highest. Additionally, at the urban site, two isomers of $C_7H_7NO_4$ were detected, likely representing methyl-nitrocatechol and nitroguaiacol (L4, Al-Naiema and Stone (2017)). Finally, SKI samples reveal a significant signal for $C_8H_6N_2O_2$, which can

be attributed to nitroindole (L4), known as a BrC-related compound from the oxidation of indole in presence of $NO_2$ (Jiang et al., 2024). The presence of this gaseous amine in the atmosphere has been attributed to many sources, such as vegetation, biomass burning, industrial activities and animal husbandry (literature within Xue et al. (2022)). We observe a similar annual pattern for nitroindole at urban (MI) and rural-agricultural (SKI) sites. However, SKI exhibits higher signal intensities, indicating a closer proximity to the precursor sources, i.e. animal husbandry. Since the DBE/#C of nitroindole is 0.875, it is

also highlighted in Figure 2 as the blue circle with coordinates (7;8). In positive ionization mode, the highest intensities for CHO(+) compounds are measured in MI as $C_{11}H_{22}O_5$ and $C_{12}H_{26}O_5$, attributed to ASOA from alkane photooxidation (Zhang et al., 2014), and more specifically to *n*-dodecane SOA (L4, Li et al. (2021)). The third highest signal at MI corresponds to the highest one at SKI, i.e. $C_{10}H_{22}O_4$, although no correspondence was found in the literature. At SKI site, $C_8H_{16}O_2$ and $C_{20}H_{26}O_3$, the second and third highest signals, are attributed to BBOA by Kong et al. (2021) and Smith et al. (2009). For

the nitrogen-containing compounds (CHNO) detected in positive mode, the formulae corresponding to the *m/z* values with the highest signals,are $C_9H_{11}NO_2$, $C_{12}H_{17}NO$ and $C_6H_{15}NO$ for MI site and $C_9H_{18}N_2O$, $C_{12}H_{23}NO$ and $C_{12}H_{25}NO$ for SKI, tentatively assigned to 1-[[(2R)-oxolan-2-yl]methyl]piperazine, a plant protection product (L4, ECHA), a compound found in agricultural residue burning (L4, Li et al. (2024)) or to n-octyl pyrrolidone (L4, ECHA, last access: June, 15$^{th}$ 2024) also used as co-solvent in crop protection formulations, and N,N-dimethyl capramide, an additive in pesticide formulations (L4, ECHA).

As these last mentioned compounds are anthropogenic chemicals, an unambiguous molecular identification can be achieved via target analysis.





## 3.2 Organic aerosol hierarchical clustering

In the following, we describe the hierarchical cluster analysis of the complete dataset. We conducted two independent cluster analyses for the two sites to ensure that the differences between the sites are maximized, enabling an unbiased assignment of a compound to a cluster regardless of its temporal behavior at the other site. It is important to clarify that the presence of a formula within a cluster does not necessarily indicate that the corresponding compound is emitted by the source after which the cluster is named. Rather, it may suggest a shared emission driver or reactive pathway that causes the compounds in the same cluster to exhibit a similar temporal pattern. However, many anthropogenic sources are known to be active year-round, with their influence often varying over time. In addition to the seasonality of emission strength, the meteorological variability can drive different reaction pathways. An increased oxidation of anthropogenic VOCs during the summer months might consequently lead to the assignment to a biogenic cluster. Even more difficult to assign to an anthropogenic cluster are SOA compounds that originate from the oxidation of volatile chemical products (VCP), such as limonene. On the other hand, such products may experience an increase in their concentration during wintertime due to a reduced mixing layer height. As such, the following discussion does not intend to definitively attribute a source to each detected compound, but rather to provide insights into their temporal trends and which other compounds show similar behavior over the course of the year and assign them to a representative molecular or sector-specific cluster. Furthermore, the use of highly resolved chromatographic and spectrometric techniques yields a large number of isomers, which cannot be precisely identified in the absence of specific reference standards of characteristic fragmentation spectra. As a result, comparisons with data from literature may also be subject to this inherent uncertainty.

At each site we examined the compounds that were detected in more than 10 % of the analyzed samples. This leads to investigate via hierarchical cluster analysis (HCA) 6,487 and 5,499 parameters for MI and SKI, respectively. We also used external data from meteorological factors (such as global radiation, daily temperature, and relative humidity), gaseous species concentrations ($NO_x$, $O_3$, benzene and toluene), ion chromatographic and thermo-optical transmittance data (i.e., EC and OC) to improve our interpretation of the results. The HCA revealed seven main clusters for the urban site (Figure S5) and five for the rural site (Figure S6). According to the temporal patterns, three clusters at both sites showed a clear increase in intensity during the colder season. This could suggest either a source activation during the winter, an amplification of the impact due to a lowering of the mixing layer height, a gas-to-particle partitioning phenomenon, or photochemical degradation during summer. In either case, this behavior suggests that the compounds in these clusters have a likely anthropogenic origin. Additionally, at MI, two clusters show an increase during the warmer season, whereas only one cluster exhibits this behavior at the rural site. The remaining clusters do not show a clear temporal pattern, which could be due to a sporadic rise or activation of the sources. The contribution of the clusters, as monthly average, is shown in Figure 3. In the Supplement, we show the fingerprints of each cluster using the retention time versus the molecular mass of each compound, the Van Krevelen diagram, and the Kroll plot to infer the cluster's chemical signature. Finally, the molecular formulae with tentative identifications and cluster attribution are listed in Additional Data (Table A1).





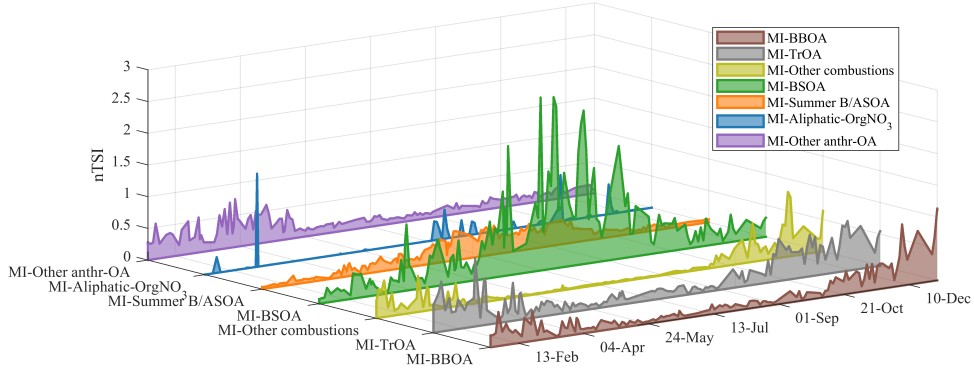

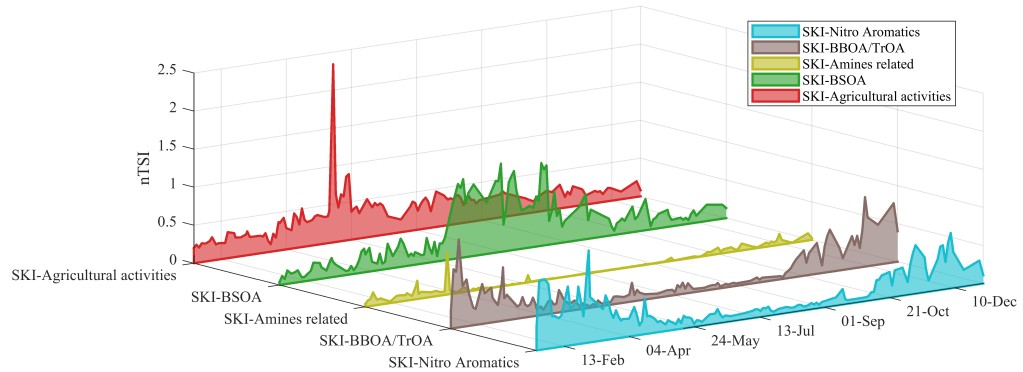

**Figure 3.** Time series of the nTSI of the clusters identified at MI (upper plot) and SKI (lower plot) sites.

### 3.2.1 Wintertime clusters

The molecular fingerprints of the wintertime clusters (Figures S7, S8, S9, S14, S15, and S16 in Supplement) are character-
ized by a significant contribution of N-containing compounds and low O/C ratio values (median<0.45). The region of the Van
Krevelen diagram that is typically associated with aromatic compounds (O/C between 0 and approximately 1 and H/C between
approximately 0.5 and 1) is densely populated. At the urban site, we identify a distinct cluster in which this region is predomi-
nantly occupied by CHO compounds (Figure S7). Notably, the nTSI of this cluster strongly correlates with soluble potassium
($R^2$ = 0.705) and levoglucosan ($R^2$ = 0.734), both established tracers of biomass burning sources (e.g., Yu et al., 2018). This
suggests that biomass burning is likely the dominant source for most compounds in this cluster. A few CHNO compounds,
such as nitrophenol ($C_6H_5NO_3$, L1), isomers of methyl-nitrophenol ($C_7H_7NO_3$, L4), methyl-nitrocatechol or nitroguaiacol
($C_7H_7NO_4$, L4), vanillin ($C_8H_8O_3$, L4) and vanillic acid ($C_8H_8O_4$, L4), also fall within the aromatic region of the Van Krev-
elen diagram of this cluster. In the CHO class, we identified phthalic acid (L1) and its isomers. Phthalic acid is released into
the atmosphere from several anthropogenic sources, such as combustion of plastics or oxidation of traffic exhaust PAHs, such



as naphtalene (e.g., Lui et al., 2023; Al-Naiema et al., 2020; Al-Naiema and Stone, 2017; Deshmukh et al., 2016; Kawamura
and Kaplan, 1987). The presence of biomass burning markers and the correlation with the external tracers suggest that BB is
the dominant source for the "MI-BBOA" cluster. Its nTSI has the highest coefficient of determination with OC concentrations
($R^2$=0.813), suggesting also that BB is a relevant source for carbonaceous matter concentrations at MI site during wintertime.
The nTSI pattern of the cluster called "MI-Other combustion" (Figure S8) shows a good correlation with soluble potassium
($R^2$=0.718) and levoglucosan ($R^2$=0.794, populated by nitrocatechol ($C_6H_5NO_4$, L1), three isomers of methyl-nitrocatechol
($C_7H_7NO_4$, L4) and coniferyl alcohol ($C_{10}H_{12}O_3$, L1). The characteristic of this cluster is mainly the presence of a large
number of CHN(+) compounds, such as $C_7H_{12}N_2$, $C_8H_{14}N_2$, $C_9H_{10}N_2$, $C_{12}H_{10}N_2$, which are formulae attributed to BBOA
by Smith et al. (2009) and Fleming et al. (2018). Moreover, even with slightly weaker correlation, this cluster indicates an
influence from traffic sources, as suggested by $NO_2$ ($R^2$=0.567) and benzene ($R^2$=0.570), leading to a cluster possibly affected
by both sources. However, we also observe a weak correlation to $EC_{traffic}$ ($R^2$=0.354), estimated by the difference between
EC and $EC_{non-fossil}$ (Daellenbach et al., 2023). This approach derives from the parametrization described in Zotter et al.
(2014), which reported a $EC_{non-fossil}$-to-levoglucosan ratio of 0.87 for stations south of the Alps and assuming that EC is
due to fossil (coal combustion and traffic exhaust) and non-fossil combustion (wood burning).

On the other hand, these external tracers correlate better with a third wintertime cluster found at the urban site ($R^2$=0.648,
0.829 and 0.747 for $EC_{traffic}$, $NO_2$ and benzene, respectively), which fingerprint is shown in Figure S9. The nTSI of this
cluster shows increasing values during the weekdays and lower ones in the weekends. Many tracers attributed to traffic exhaust
in the literature are found here: e.g. $C_4H_6O_4$ (succinic acid, L4, Lui et al. (2023)), $C_7H_{12}O_7$ (L4, Thoma et al. (2022)), and
$C_9H_8O_4$ (methylphthalic acid, L4, Ikemori et al. (2021)). In addition, N-containing features enrich the so-called "MI-TrOA"
(Traffic OA) cluster, including aliphatic amines such as $C_8H_{18}N$, $C_7H_{17}N$, $C_9H_{21}N$ (as reported in Cao et al. (2023)). This
cluster contains probable cyclic compounds, like $C_{10}H_{14}N_2$, which could be nicotine (L4) and nitro-aromatic compounds,
$C_6H_3N_3O_7$ (picric acid, L4, Lin et al. (2015)). More in general, it includes N-organic containing compounds (NOCs), some
reported in the literature from traffic exhaust, like $C_9H_{19}NO$ (N,N-dibutylformamide, Rogge et al. (1993), L4). These characteristics clearly distinguish this cluster from MI-BBOA and MI-Other Combustions clusters discussed earlier. In addition, we
found some CHOP compounds, such as $C_{18}H_{39}O_7P$ (tris(2-butoxyethyl)-phosphate, L4, Ungeheuer et al. (2021)), $C_{18}H_{15}OP$
(triphenylphosphine oxide, L4, Faiz et al. (2018)), and $C_8H_{19}O_5P$ (found in Lao et al. (2022)), attributable to flame retardants.
This class of compounds are often observed as non-combustion emission by vehicles too (Dong et al., 2025). Within the wintertime clusters, 96 out of the 239 compounds likely HOA are observed, based on their values of $OS_c$ (between -2 and -1) and
#C (between 18 and 30) (Kroll et al., 2011), distributed as 16 %, 30 % and 54 % among BBOA, Other Combustion and TrOA
clusters, respectively. As an example, $C_{19}H_{26}O_2$ and $C_{20}H_{28}O_3$ are attributed to BBOA (Zhang et al. (2023a) and Smith et al.
(2009), L4), whereas $C_{18}H_{28}O$ to vehicular emissions (Li et al. (2021), L4).
At the rural site, a total of 82 HOA compounds are found, of which 46 appear in one of the three clusters with a winter temporal pattern. The cluster named "SKI-BBOA/TrOA" (Figure S14) exhibits a strong correlation with both biomass burning
(i.e. soluble potassium, $R^2$ = 0.674, levoglucosan, $R^2$ = 0.708) and traffic (i.e. benzene, $R^2$ = 0.769, $EC_{traffic}$, $R^2$ = 0.539,
and $NO_2$, $R^2$ = 0.678) markers. Notable compounds include vanillic acid ($C_8H_8O_8$, L4, Simoneit (2002)), nitrophenol (L1),





dodecanitrile ($C_{12}H_{23}N$, L4, Simoneit et al. (2003)), nitrocatechol (L1), nitrosalicylic acid ($C_7H_5NO_5$, L4, He et al. (2022b)),

acetosyringone ($C_{10}H_{12}O_4$, L1, Li et al. (2020b) and references therein), phthalic acid (L1), methylphthalic acid (L4), and 4-hydroxy-3-nitrobenzyl alcohol ($C_7H_7NO_4$, L4, Al-Naiema and Stone (2017)). Also, 33 HOA compounds fall in this cluster, such as $C_{20}H_{28}O_3$, and $C_{20}H_{26}O_2$, attributed by Smith et al. (2009) to BB sources.

Similar to clusters identified in the MI dataset (Figure S7 and S9), S-containing compounds were also detected in wintertime clusters. Among these, we found $C_{12}H_{20}O_7S$ (L4, literature in Srivastava et al. (2022)), $C_8H_{18}O_4S$ (L4, Blair et al. (2017)),

both linked to SOA from traffic emissions or BSOA. These last include both polar compounds like $C_5H_8O_7S$ (L4, SOA from isoprene and $SO_2$, Claeys and Maenhaut (2021)) or $C_4H_6O_7S$ (L4, SOA from isoprene and $SO_2$, Lin et al. (2022)), as well as larger compounds that were retained more effectively by the chromatographic column, such as $C_{10}H_{17}NO_7S$ (L4, Surratt et al. (2008)). The temporal pattern of these wintertime S-containing compounds suggests the importance of the co-presence of their biogenic precursors and high NOx concentrations. At the SKI site, wintertime S-containing compounds are equally

distributed between SKI-BBOA/TrOA and the second cluster with a winter temporal pattern (Figure S16). We observed a moderate correlation with inorganic nitrate ($R^2$=0.541) and ammonium ($R^2$=0.415) ion concentrations, known for their significant contribution to $PM_{2.5}$ during winter months in the Po Valley (Ferrero et al. (2019)), EC ($R^2$=0.669, $R^2_{EC-traffic}$=0.524) and benzene ($R^2$=0.515). On the other hand, the cluster fingerprint clearly shows a strong contribution of nitro-aromatics compounds, such as nitrophenol and nitrocatechol, indicating a probable combustion source (BB) contributing to OA with

reaction products sensitive to the same air conditions for ammonium nitrate gas-to-particle partitioning, and OC concentrations ($R^2$=0.681). Because of the strong presence of nitro-aromatics, we labeled this cluster "SKI-Nitro Aromatics". Within this cluster, 11 P-containing compounds significantly contribute to the CHOP family's nTSI, as shown in Figure 1. Among these, $C_6H_{15}O_4P$, $C_2H_7O_4P$, and $C_4H_{11}O_4P$, show the highest nSI. The molecular formulae match with triethylphosphate, used as a flame retardant and as an additive in plant protection formulations, dimethylphosphate, and diethyl hydrogen phosphate,

likely degradation products of other organophosphates (Liu et al., 2021).

The third wintertime cluster detected at SKI (Figure S15) contains many compounds assigned to "MI-Other combustion". This SKI cluster primarily consist of NOCs (approximately 90 % of the features), mainly detected in positive ionization mode. This suggests a prevalence of reduced compounds, likely amine-related CHN(+). The $X_c$ values below 2.5 for 65 % (median 2.3) of the features indicate that these compounds are mainly aliphatic. CHNO(+) compounds exhibit low O/C (0.17±0.04) and

O/N (1.2±0.4) ratios, consistent with the presence of amino groups and O-containing substituents (average carbon number 8.11±0.57). The peak abundance-weighted average of the nitrogen number in the CHN family is 1.98±0.14. Based on these observations, we labeled this cluster as "SKI-Amines related". The time series of this cluster reveals certain samples with high nTSI values for all the compounds it contains. This behavior is not observed for compounds in other clusters. The Kendrick mass defect analysis highlights homologous series such as $C_nH_{2n}N_2$ (n=7, 9, 11), $C_nH_{2n-2}N_2$ (n=5-13), $C_nH_{2n}N_2$ (n=6, 8,

10, 12). These series suggest probable methylation/demethylation processes in their genesis. However, previous studies have shown that the presence of reduced NOCs in the atmosphere can stem from a variety of sources. Murphy et al. (2007) reported that cooking, industrial sources, biomass combustion, and traffic are just a few of the possible sources, emphasizing that livestock activities are the predominant contributors of these compounds. In this context, it is worth to notice that other authors





(such as Silva et al. (2008), and Malloy et al. (2009)) have demonstrated that the reaction of aliphatic amines with nitrate
radicals leads to the formation of imines in the aerosol phase, while Smith et al. (2021) have described the ability of imines to
interact with light, suggesting their role as a component of secondary BrC.

### 3.2.2 Summertime clusters

At both sites we detect highly populated cluster peaking during summertime, with 1,487 and 1,623 compounds for MI and SKI,
respectively, mostly detected in negative mode. The molecular fingerprints of these two clusters (Figures S10 and S17) are very
similar. Summertime clusters are mainly constituted by aliphatic CHO and CHOS compounds, with short carbon chains (peak-
abundance weighted average, #C=6.7±0.7). Interestingly, the values of the averaged carbon number (weighted by the peak
area) increase during the summertime more than the averaged oxygen number, leading to lower O/C ratios. Because of this,
the average $OS_c$ decreases as well in the warmer season. Compared to the other summertime clusters, these clusters exhibit the
highest O/C ratio. CHO compounds have H/C ratios ranging from 1 to 2 and O/C ratios between 0.3 and 1.5. The high oxygen
atom content (median = 6.0) observed in the S-containing compounds suggests that they are organosulfates. In this regard,
many formulae match with ones we found in literature from isoprene oxidation products in presence of sulfuric acid, such as
$C_4H_8O_7S$, $C_4H_8O_6S$ (likely 2-methylglyceric acid organosulfate, L4, Claeys and Maenhaut (2021)), $C_5H_{12}O_7S$ (SOA from
2-methyltetrol organosulfate, L4, Lin et al. (2022)) and $C_5H_{10}O_7S$ (L4, Nestorowicz et al. (2018)), and from monoterpenes
like $C_8H_{14}O_7S$, $C_7H_{12}O_6S$, $C_{10}H_{18}O_7S$ (L4, literature within Chen et al. (2018); Jiang et al. (2024)). The two clusters in-
clude MBTCA (L1), many isomers of $C_8H_{14}O_4$, which could be for example caric acid (L4, Gómez-González et al. (2012)),
SOA from limonene ($C_7H_{10}O_5$, L4), $\beta$-pinene ($C_8H_{14}O_5$) and $\Delta^3$-carene ($C_6H_{10}O_2$), also found by Thoma et al. (2022)
using HESI(–). Based on these indicators, we conclude that the two clusters at MI and SKI represent summertime biogenic
SOA ("BSOA"). Alongside the CHO and CHOS compounds, a considerable amount (that are 205 and 104 compounds, which
represent 13.8 % and 6.5 % of the total compounds in BSOA cluster for MI and SKI, respectively), even with lower nTSI, are
N-containing compounds detected in positive ion mode. Many of these were also listed in Zhang et al. (2023a), and attributed
to wood burning (L4). Although the identification level is only 4 (unequivocal molecular formula), we hypothesize that these
compounds are rising during summertime from oxidized BBOA, because of the meteorological and climatic conditions, such
as the higher concentrations of oxidants species ($O_3$, $NO_3$ and OH radicals) induced by higher solar activity in this season.
Indeed, the moderate correlation between NOCs for MI-BSOA and daily average $O_3$ concentration ($R^2$=0.583) provides an
insight of this.

This hypothesis is further supported by the dominance of the CHNO(+) family contributing to the nTSI of the second summer-
time cluster identified at the urban site (Figure S11). The whole group is mainly populated by compounds detected in positive
ion mode and belonging to CHO and CHNO families (329 and 225 features, respectively). Both exhibit higher correlation
with daily average $O_3$ ($R^2$=0.576 and 0.458, respectively) and global radiation (0.434 and 0.381) values in respect to the same
families belonging to MI-BSOA (0.414 and 0.366 with $O_3$, 0.212 and 0.190 with the global radiation). While the cluster's H/C
ratio is close to the one of MI-BSOA (median 1.63 vs 1.56, respectively), its O/C ratio is significantly lower (0.37 vs 0.60),





resulting in a lower averaged $OS_c$ (-0.9 vs -0.3). In this cluster we also found many features attributed to biogenic sources, such as $C_7H_{10}O_4$, $C_8H_{11}NO_7$, $C_{10}H_{15}NO_9$ attributed to SOA from monoterpenes (L4, Finessi et al. (2014), Faxon et al. (2018),

Pullinen et al. (2020)), and falling in the SKI-BSOA cluster too. However, more than 40 % are features not detected at SKI, of which 70 % are detected in positive mode, like $C_9H_{20}O_4$ and $C_{13}H_{24}O_4$, which instead are found in Qi et al. (2019) in an urban site in Zurich and apportioned to cigarette smoke OA and cooking OA, respectively. Because of this strong mix of both biogenic and anthropogenic source assignments, we labeled this cluster as "MI-Summer B/ASOA". Due to the large number of undetected compounds in the SKI-BSOA and in other clusters, the temporal pattern and the mix of both biogenic and an-

thropogenic SOA, we can only speculate that this cluster has a contribution due to VCP SOAs. Supporting the hypothesis of SOA from VCPs, we find in addition 29 compounds attributable to the HOA class of which only 8 were detected at the rural site. The BSOA clusters from both sites share only one HOA compound ($C_{18}H_{29}NO_3$). However, at SKI, 11 out of 17 HOA compounds were unknown at the urban site. Hence, HOAs contribute significantly to the chemical variability between the two locations.


### 3.2.3 Agriculture and remaining clusters

HCA revealed additional clusters, which do not exhibit a distinct seasonal pattern. At the rural site we identified a cluster, which display a strong increase during the spring season, with a largest values in May. This is mainly due to a group of molecules belonging to CHNO family and that is responsible for the increase in the overall nTSI of this family shown in Figure 1. Among

these we found $C_{12}H_{25}NO$, which matches with N,N-dimethyl capramide, used as an additive in pesticide formulations (L4, ECHA, last access: June, $15^{th}$ 2024), the aforementioned $C_{12}H_{23}NO$, attributed to n-octyl pyrrolidone, used in products as solvent for pesticides (L4, ECHA), $C_9H_{18}N_2O$, attributed to 1-[[(2R)-oxolan-2-yl]methyl]piperazine (L4), another plant protection agent. By means of standard comparison, we confirmed the presence in SKI samples and the inclusion in this cluster of Metalaxyl ($C_{15}H_{21}NO_4$, L1, fungicide), Prothioconazole ($C_{14}H_{15}Cl_2N_3O$, L1, fungicide), Metolachlor ($C_{15}H_{22}ClNO_2$, L1,

herbicide), Pyrimethanil ($C_{12}H_{13}N_3$, L1, fungicide), Cyprodinil ($C_{14}H_{15}N_3$, L1, fungicide), Clomazone ($C_{12}H_{14}ClNO_2$, L1, herbicide), Terbuthylazine ($C_9H_{16}ClN_5$, L1, herbicide), Dimethenamid ($C_{12}H_{18}ClNO_2S$, L1, herbicide), and Tebuconazole ($C_{16}H_{22}ClN_3O$, L1, fungicide). Although some compounds are from verified biogenic (such as pinic and pinonic acids, L1) or probably from other sources (such as $C_{15}H_{24}O_4$ and $C_{12}H_{23}NO$, from agricultural residue burning, L4, Zhang et al. (2023a)), we labeled this cluster "SKI-Agricultural activities", pointing out the exclusiveness of many of these features found at SKI. In

addition, the cluster fingerprint (Figure S18) is enriched by S-containing compounds. In this context, Blair et al. (2017) found a number of organosulfates from photooxidation of hydrocarbons from diesel fuel in presence of $SO_2$. Many of the molecular formulae the authors have found in their chamber experiments were found at both MI and SKI sites, mainly attributed to BSOA clusters, likely due to the increasing availability of $H_2SO_4$ during summertime. Nevertheless, most of the S-containing compounds attributed to SKI-Agricultural activities are not detected at MI site. Thus, we speculate that isomers with these formula,

such as $C_4H_8O_8S$, $C_8H_{16}O_8S$, $C_{16}H_{26}O_3S$, $C_7H_{12}O_7S$ (L4), could be attributed to agricultural tractors emissions. Out of a total of 1,398 features that make up the cluster SKI-Agricultural activities, as many as 645 features are completely absent at





the Milan site, of which more than 64 % are detected in positive mode.

At the MI site, we isolated a cluster populated by CHNO compounds only, mainly measured in positive mode. These features show low nTSI variability along the year with the exception of few events, which show an increase in intensity by orders of magnitude. The fingerprint (Figure S12) displays that the compounds forming this cluster fall in a very small area of the Van Krevelen diagram (O/C<0.5 and 1.4<H/C<2.1). Moreover, observing the relation between the chromatographic retention time and the molecular mass, we infer that a subset of these features is likely due to dimers from high-NOx terpene oxidation Thoma et al. (2025). The median #C (16) value is the highest among the clusters found at MI, while the median $OS_c$ (-1.3) value is the lowest: this is due to the high amount of HOAs, which account for 39 % of the features which belong to this cluster. Very few formulae match with ones found in literature: some of them were attributed to cigarette smoke and cooking by Qi et al. (2019) and to agriculture residue burning by Zhang et al. (2023a). Finally, due to the area span in the Van Krevelen space, and the O/N ratio above 3 (3.1±1.4, average±standard deviation), we labeled this cluster "MI-Aliphatic-OrgNO$_3$". The seventh cluster found in MI exhibits a sharp decrease in nTSI in mid-May 2021, stabilizing at nearly constant levels for the remainder of the year. Compounds detected in positive mode outnumber those detected in negative mode. The cluster fingerprint (Figure S13) shows compounds mainly populating the Van Krevelen space of aliphatics, with a O/C ratio close to 0.5 and an O/N ratio (for CHNO) of 2.7±0.6, which does not fully restrict the possibilities of N-containing functional groups other than nitrate ester ones. Marker compound matches do not strongly support any hypothesis about the origin of these molecules. We found p-coumaric acid ($C_9H_8O_3$, L1), which is known to be a product from biomass burning, especially, but not only, from grasses one (Oros et al., 2006). Moreover, $C_{11}H_{22}O_5$ was found in Li et al. (2021) (L4) during oxidation of *n*-dodecane experiments and $C_7H_{13}NO_2$ was detected as *m*-xylene SOA (L4, Li et al. 2018). Due to the sharp decrease of the nTSI we can speculate that the driving source of this cluster is an anthropogenic activity but no clear hint is found. Thus, we labeled this cluster "Other anthr-OA".

### 3.3 Organic aerosol molecular variability between Milan and Schivenoglia

The use of the HCA technique allowed us to characterize cluster of compounds with similar temporal pattern. Between the two sampling sites, similarities and differences in molecular composition of the interpreted clusters are discussed in 3.2. Their contribution to the annual nTSI is also diverse: winter-peaking clusters in Milan, linked to traffic, biomass burning, and a mix of both, account for about 16.1 %, 10.9 % and 8.1 % to annual nTSI, respectively. At Schivenoglia, mixed traffic and biomass burning sources account for 14.9 %, the cluster linked to biomass burning with ammonium nitrate formation account for 20.2 %, and the third cluster containing aliphatic amines possibly from agricultural reside burning, accounts for 2.6 % to the annual nTSI. The compounds attributed to biogenic-related sources, peaking during summertime, have a higher impact at MI (31.4 %) rather than at SKI (26.4 %): in addition, a fraction of MI-Summer B/ASOA cluster (11.0 %) might also be due to biogenic sources or stem from VCP (e.g. limonene) oxidation. At the urban site, anthropogenic sources have an additional 14.1 % contribution to the annual nTSI due to MI-other anthr-OA cluster, and 2.1 % due to MI-Aliphatic OrgNO$_3$ one, even though no clear attribution is performed. At the rural site, anthropogenic activities linked to agricultural practices are clearly

49151



observed and quantified to 27.8 %.

Notably, some clusters of MI and SKI show a similar molecular composition. For example, 61 % of compounds clustered in MI-BSOA are in common with SKI-BSOA, and in turn SKI-BSOA shares 56 % of its compounds with MI-BSOA and 12 % with MI-Summer B/ASOA. Both biogenic clusters show a number of compounds that are not classified at the other site: at
MI 350 compounds are not detected (or detected in less than 10 % of the analyzed samples) in SKI, whereas a higher amount (436) are observed at the rural site but not at the urban one. This provides a first insight of the molecular variability between the two environments. For SKI, the cluster that contains the largest number of these site-specific compounds is SKI-Agricultural activities (645, i.e. 46 % of compounds falling into it). On the contrary, every cluster found in MI contains a comparable amount of site-specific compounds. The chord chart (adapted from Liu (2025)) in Figure 4, graphically assists the assessment
of the site differences, where the width of the flow-lines is proportional to the number of compounds that are common between the two connected clusters. The chart suggests that combustion sources have a different profile at the rural compared to the urban site. MI-BBOA, MI-TrOA and MI-Other combustion share 45 % (n=412), 19 % (n=175), and 33 % (n=306) compounds only with SKI-BBOA/TrOA, and 11 %, 30 % and 15 % with SKI-Nitro Aromatics, respectively. This leads to a percentage of site-specific combustion-related compounds from this MI clusters of 36 %, 40 %, and 30 % respectively. We found a total of
2,513 and 1,525 compounds that are site-specific for MI and SKI, respectively, confirming the higher molecular-diversity at the urban site. The site-specific fingerprints, shown in Figures S19 and S20, highlight that the urban site is richer in unpolar compounds with high molecular mass, and in CHO-containing aromatics. Among aliphatics, N-containing compounds are the most prominent compounds present at the urban site. Figure 5 illustrates the contribution to the monthly nTSI of each cluster at both sites. As expected, during wintertime the nTSI explained by the site-specific compounds (in darker colors) is lower
than in the other seasons. The reason is explainable by the lack of both weak oxidant conditions, leading to a longer lifetime of organic molecules, and meteorological events able to abate PM concentrations: as a result, aerosols disperse, mix and age in the lower layers of the troposphere in a vast area of the Po basin. Limiting the discussion on the site-specific features only, during winter in both sampling sites the anthropogenic clusters explain between 10 % and 15 % of the nTSI, whereas site-specific BSOA compounds are almost negligible. On the opposite, from May to September this cluster provides the highest molecular
variability at MI in respect of SKI. The rural site, in fact, exhibits site-specific compound profiles predominantly originating from agricultural activities throughout the year, with only modest contributions from the SKI-Nitro-Aromatics cluster in winter and from source SKI-BSOA in summer. Between May and September, both sampling sites exhibit a significant molecular variability, which rises from 15 % to 25 %. It is worth noting that, at both sites, the lowest concentrations of OC were measured in May ($2.0 \pm 0.6$ $\mu$g/m$^3$ and $1.6 \pm 0.3$ $\mu$g/m$^3$, Figure 1). At the same time, the highest plant protection product intensities
at the SKI site were detected in May, suggesting that, despite theoretically indicating a reduced health impact that month, the aerosols could have a non-negligible toxicological effect during that period at SKI. Simultaneously, the MI-Summer B/ASOA cluster, predominantly composed of compounds detected in positive mode, with a significant fraction unknown at the SKI site, shows its highest contribution to nTSI (7 % accounting for site-specifice compounds only) once again in May. These results highlight the need to assess the health effects of aerosols not only based on the mass quantification metrics but also, and perhaps
most importantly, based on chemical composition and evaluate health effects of emissions from agricultural practices.





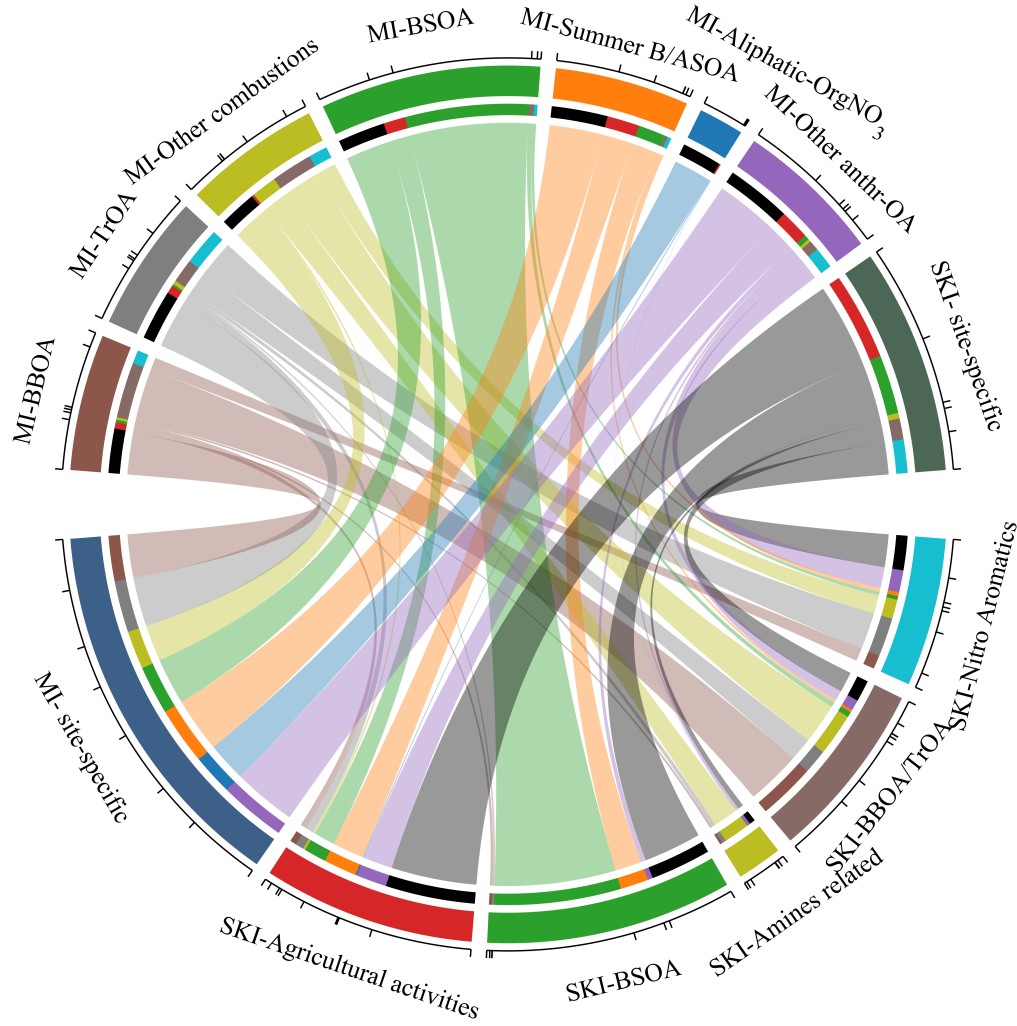

**Figure 4.** Chord chart illustrating, for each cluster, the number of compounds shared between the two sites, as well as the number of cluster-specific compounds unassigned at the other site, which are considered site-specific.

Finally, we select sample pairs with similar OC concentrations (within a tolerance of 2.0 $\mu$g m$^{-3}$) and contribution to PM$_{2.5}$ concentration (within a tolerance of 5 %) collected on the same day at both sites (hereinafter referred as "similar samples"). These conditions lead us to select 54 sample pairs across all months but June and December 2021. The differences of the nSI highlight the dissimilarities between sample pairs of similar OC concentrations. Once again, we take advantage from the HCA

performed (Section 3.2) and classify the enrichment in compounds based on their source attribution. The bar chart in Figure 6 shows the compound membership clusters with residuals, averaged on a monthly scale, favorable to MI (upper semi-axis) or SKI (lower semi-axis) for the selected samples as contribution to nTSI (see Supplement, Section 1.1 "Residuals contribution calculation"). Both semi-axes are plotted as absolute values to display only the magnitude of the residuals. At MI site, the



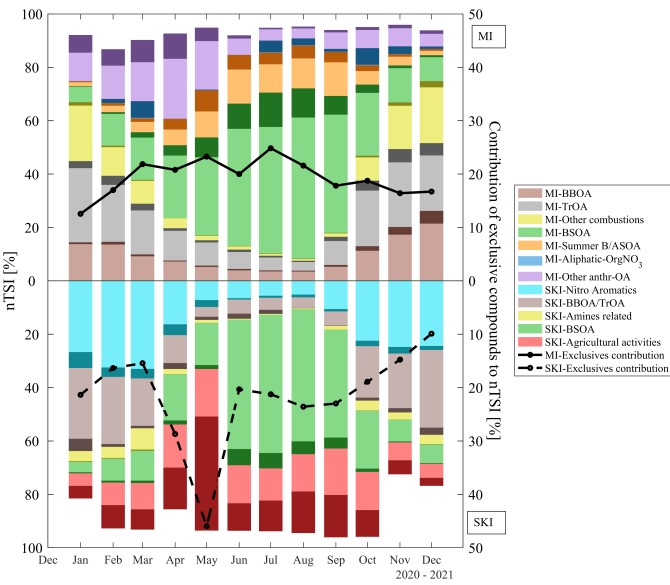

**Figure 5.** Contribution to the normalized total signal intensity (nTSI) of the classified compounds divided by cluster attribution. Darker colors represent the fraction of site-specific features of each cluster.

molecular variability is mostly due to compounds detected at both sites (lighter colors) for the majority of the sources with

a seasonal pattern. The residuals of common compounds contribute about 38±7 % to nTSI (reported as an annual average), whereas the residuals of site-specific compounds contribute 18±5 %. At SKI residual contributions of common compounds account to 26±6 %, slightly higher than for site-specific ones (21±11 %). In particular, Figure 6 shows that the agricultural activities drive both the site-specific features and the high residual contributions. Moreover, the highest differences are observed in May, when the OC concentrations are the lowest. This reinforces the assertion that the sources at the two sites have distinct

emission profiles and formation pathways.

## 4 Atmospheric implications on light-absorbing aerosols

In the initial dataset, we found 4,370 compounds with DBE/#C ranging between 0.5 and 0.9 (Figure 2), of which 1,551 and 1,247 were classified for MI and SKI, respectively, using HCA. Among these, we detect $C_8H_7NO_5$ attributed to 3-methoxy-

4-nitrobenzoic acid (L4, Li et al. (2020a)), $C_7H_6O_2$, attributed to benzoic acid or p-hydroxybenzalehyde (L4, Worton et al. (2011)), p-coumaric acid, nitrophenol, nitrocatechol, acetosyringone, phtalic acid, $C_{10}H_{10}O_3$ assigned to coniferyl aldehyde (L4, Oros et al. (2006) and Huang et al. (2022)), $C_6H_6N_2O_3$ attributed to amino-nitrophenol (L4, literature within Huang et al. (2024)), $C_6H_3N_3O_7$ attributed to picric acid (L4, Lin et al. (2015)), $C_8H_6N_2O_2$, likely nitroindole (L4). Potential BrC compounds are listed as Additional data (Table A2) together with their cluster attributions and the average percentage



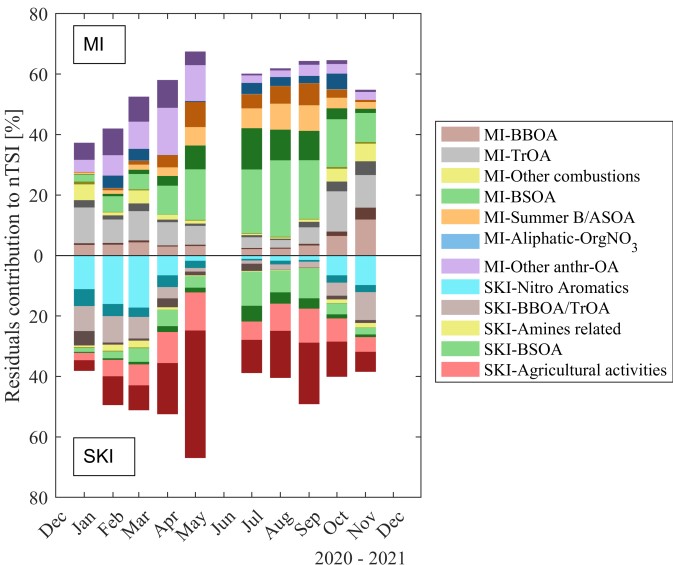

**Figure 6.** Cluster attribution of the contributions to the nTSI from residuals, limited to pairs of samples that show similar OC concentrations and OC contribution to PM$_{2.5}$. Darker colors indicate the fractional contribution from compounds that are site-specific to each sample.

contribution to nTSI. The fingerprints of the classified BrC features at MI and SKI are shown in Figure S21 and S22. The contribution of classified BrC-related compound to nTSI accounts for about 6 % at both sites during summer and increases up to 38 % during winter, with an annual average of 14 % at MI and 13 % at SKI, respectively. Most of BrC-related compounds fall into combustion clusters (Figure S23): at the urban site, we observe that all BrC-combustive clusters exhibit a stronger contribution during wintertime, with the highest one due to the MI-Other combustion cluster, which is mainly populated by

aliphatic-CHNO and CHN(+) compounds. MI-TrOA cluster seems to have less impact compared to MI-BBOA, which still shows the greatest contribution for combustion sources during the summer months. At the rural site, however, the largest contribution comes from the "SKI-NitroAromatics" cluster, which is also linked to combustion but, unlike at MI site, appears to have an increasing impact on nTSI until early spring. Here too, SKI-BBOA/TrOA contributes to the presence of potentially BrC compounds throughout the year, though with a smaller contribution relative to nTSI when compared to the role of the

respective clusters found at MI.

Considering the 54 sample pairs with very similar OC concentrations and contributions to PM$_{2.5}$ mass (Section 3.2.3), Figure 7 present nested annular plots illustrating the residual contributions of all (outer circle) and BrC-related compounds only (inner circle), averaged over the entire year. The plots confirm, as expected, that combustion sources are the dominant contributors to the annual variability of BrC, accounting for nearly 75 % of the residual contributions at both sites. However, the average

contribution of site-specific BrC-related compounds is notably lower but still relevant: they constitute 23 % of the residuals at the urban site, compared to 16 % at the rural site. During the year the residual contribution of the clusters to nTSI does not vary substantially (between 2 % and 9 % for all classified BrC-related compounds, as shown in Figure 8, secondary vertical axis) at





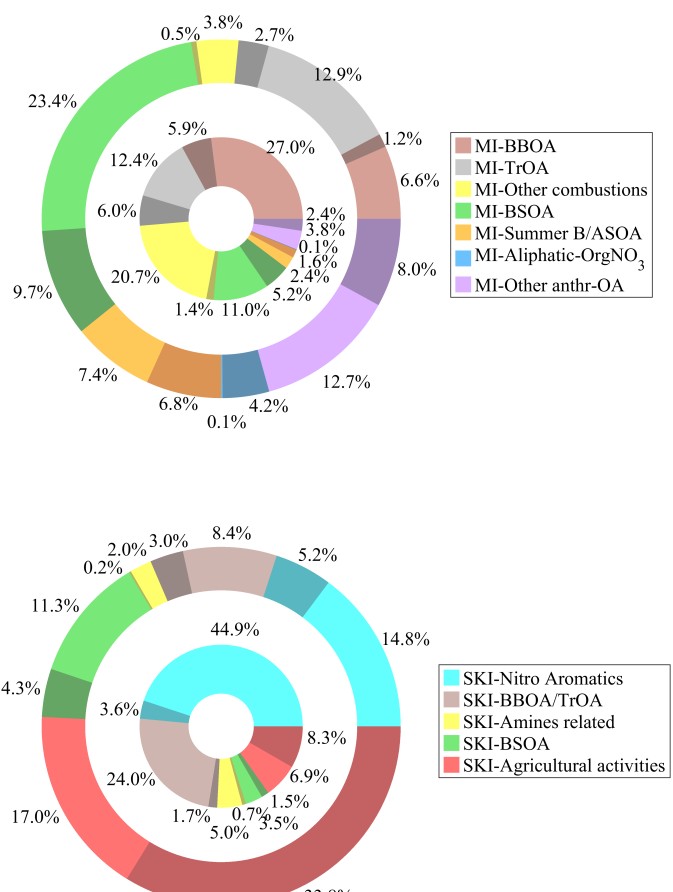

**Figure 7.** Residuals calculated on samples with similar OC concentrations and contributions to PM$_{2.5}$ for MI (upper plot) and SKI (lower plot). The outer circle illustrates the distribution of annual average residuals across all compounds attributed to sources identified at the reference site. In contrast, the inner circle focuses on the same distribution but is limited to compounds potentially classified as LAA. Darker shades highlight the contribution of the site-specific compounds.

both sites, even with a seasonality. However, because differences are observed both in abundance (residuals are not null) and in the molecular species present, a different interaction with light can be hypothesized, even under similar OC concentration

conditions.

In this regards, the AAE of eight sample pairs collected at MI and SKI sites provides an insight of this (Figure S24). For the majority of samples, the AAEs estimated for MI samples were higher than those collected at SKI: the median absolute differences were 1.40, 0.23, and 1.22 in the 230–250 nm, 250–400 nm, and 400–650 nm ranges, respectively, with absolute maximum up to 4.90. Although more measurements are needed, these results confirm that, even with a low OC concentration

variability, the molecular differences between two sites lead to a different light-absorption behavior.



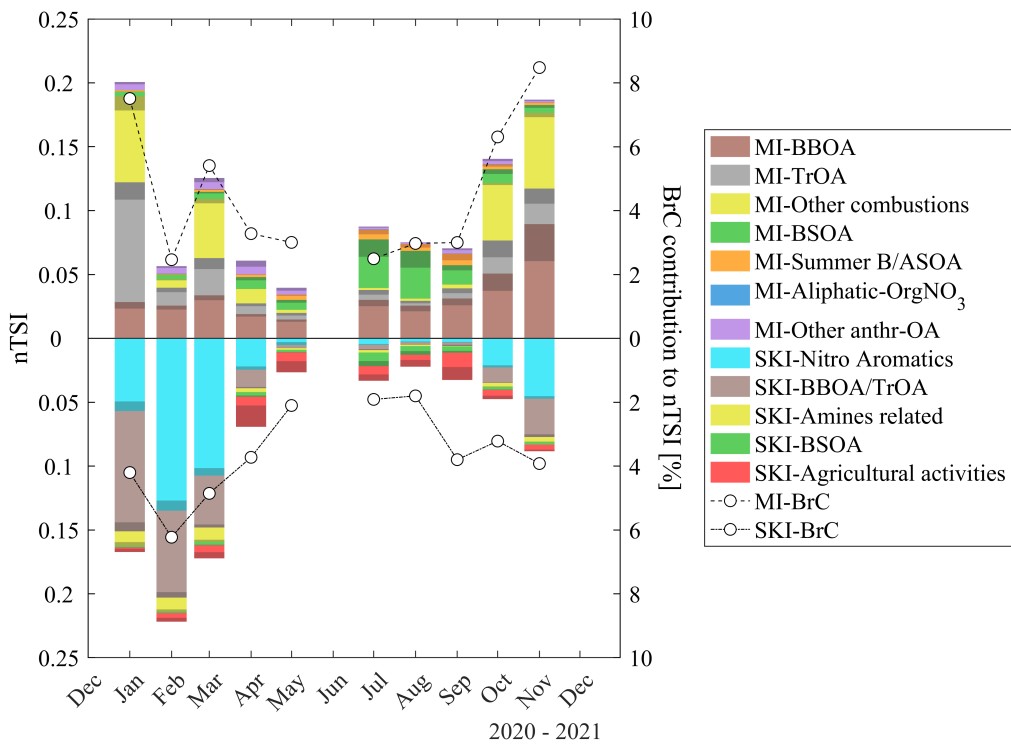

**Figure 8.** BrC residuals at MI (top) and SKI (bottom) for samples with similar OC concentrations and contribution to PM$_{2.5}$. Darker colors highlight site-specific BrC features. The secondary vertical axes show the monthly averaged contribution of the residual BrC-related compounds to the nTSI of the selected samples.

## 5  Conclusions

This study explores the chemical variability of organic aerosol in a European hotspot for air quality, such as the Po Valley. Historical data indicate that PM$_{2.5}$ and OC concentrations are often similar across sampling sites, suggesting little variability in aerosol composition. To test this, PM$_{2.5}$ samples collected in 2021 at a rural agricultural and an urban background site were analyzed. Using a hierarchical clustering approach on UHPLC-HESI-UHRMS data, compounds were grouped by known source markers and temporal patterns. At the urban site (Milan), seven clusters emerge: three peak in winter (linked to traffic combustion, biomass burning, and a mix of both), two in summer (one due to biogenic secondary organic compounds, the other made of biogenic and anthropogenic). The remaining two clusters without a clear seasonal trend likely represent unidentified anthropogenic activities. At the agricultural site (Schivenoglia), five clusters are identified, three of which peak in winter. One correlates with soluble potassium, levoglucosan, benzene, and NO$_2$, suggesting combustion sources; another links biomass burning with ammonium nitrate formation, and the third contains aliphatic amines possibly from agricultural residue burning. A biogenic SOA cluster peaks as expected during summer, and one cluster clearly emerges in late spring tied to agricultural activities, as its main compounds are plant protection products.

Focusing only on site-specific compounds of each site, the molecular richness of the urban site amounts to about 38 % of the
total classified features, contributing between 11 % and 23 % to the nTSI on monthly basis. The agricultural site is charac-
terized by 27 % site-specific features, contributing between 7 % and 23 % to the nTSI, with a peak of up to 55 % in May
due to molecules associated with plant protection products. At both sites, the lowest contribution from molecular variability is
observed in the winter months, suggesting that poor ventilation of the basin due to meteorological conditions results in a more
homogeneous Po Valley during this season.

The results indicate that, while the two sites may appear quantitatively similar, they exhibit significant qualitative differences.
Although limitations in the analytical approach may influence the findings, the observed variability suggests that it could dis-
tinctly affect both the toxicity of organic aerosols and their interactions with radiation. In this regards, we demonstrate that
$PM_{2.5}$ extracts have significant spectral behavior differences. This work encourages the aerosol research community to deepen
the molecular-level understanding of organic aerosols and to identify relationships with larger-scale effects, thereby guiding
policy makers toward the adoption of more effective measures.

Finally, the use of high-resolution and high-accuracy mass spectrometry enables the detection and identification of specific
compounds based on targeted hypotheses and data contextualization. This approach allowed us to identify the contributions
of certain sources, such as agricultural activities, which are often overlooked in traditional source apportionment methods.
Although our analysis cannot be quantitative, it provides a more refined understanding of the contributions of various anthro-
pogenic sources.

*Data availability.* The data in the graphs, as well as the molecular formulas attributed to each cluster and related to BrC, are available on
Zenodo (10.5281/zenodo.16311974). Data are available upon request to the corresponding authors.

*Author contributions.* LD analyzed the filters by HPLC-HESI-HRMS and PDA, performed the data analysis, interpreted the results, and
drafted the paper. ALV, FU, JM, LF provided suggestions for data analysis, interpretation, and discussion and edited the manuscript. EC
analyzed the filters by thermo-optical instrument and ion chromatography. CC, EC, BB, UD gave general advices and comments for this
paper.

*Competing interests.* The authors declare no competing interests.

*Acknowledgements.* The authors thank the staff at ARPA Lombardia for providing the aerosol samples. We thank Georg Menzel for filter
extraction laboratory work.



615 *Financial support.*

This project (grant no. 410009325) has been supported by the Deutsche Forschungsgemeinschaft (DFG; German Research Foundation).



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
