# Peer review of "Contrasting organic aerosol molecular composition between the urban and agricultural environment of the Po Valley"

_EGUsphere, 2025_

## Referee Comment (RC1)

Referee Report on

**Contrasting organic aerosol molecular composition between the**

**urban and agricultural environment of the Po Valley**

This manuscript investigates the molecular-level composition of organic aerosol (OA) at an urban and a rural agricultural site in the Po Valley during 2021, using ultra-high performance liquid chromatography with soft-ionization ultra-high-resolution mass spectrometry combined with hierarchical clustering analysis. The authors analyze a large number of $PM_{2.5}$ filter samples over a full year and attempt to attribute OA sources based on temporal behavior and molecular characteristics. The study provides valuable insights into the qualitative differences in OA composition between urban and rural environments, highlighting the significant influence of underappreciated sources like agricultural pesticides.

Overall, the dataset is extensive, and the analytical effort is substantial. The year-long comparison between urban and rural environments at molecular resolution is valuable and timely. The manuscript has the potential to provide important insights into OA variability in the Po Valley. However, despite the large workload and comprehensive dataset, I have significant concerns regarding data interpretation, source attribution, and the logical consistency of several discussions. These issues should be carefully addressed before the manuscript can be considered for publication.

1. Lines 35-70: The introduction first emphasizes the importance of molecular-level characterization of OA and then lists previous studies conducted in the Po Valley, but it lacks a concise synthesis of existing knowledge. I suggest adding a short summary outlining the major OA sources identified in the Po Valley, their seasonal and spatial variability, and the main hypotheses proposed to explain these variations, which would better contextualize the novelty of this work.

2. Lines 37-39: "Recently, Thoma et al. (2025) have observed that even at a rural site, OA system is composed of several thousands of compounds affected by seasonality and short- and long-transport events, with biogenic secondary OA (BSOA) representing about 70 % of compounds and 30 % attributed to anthropogenic SOA (ASOA)."
It is unclear whether these percentages refer to the number of detected compounds or to signal intensity. If based on signal intensity, this should be explicitly stated and discussed with caution, as ESI measurements are strongly affected by ionization efficiency and matrix effects, making direct interpretation of intensity fractions as SOA contributions uncertain.

3. Lines 81-82: "Non-targeted analysis allows for inferring the overall properties of OA and has the potential to identify relationships between groups of compounds and their sources or other endpoints."

This statement requires literature support and further explanation, as non-target analysis provides comprehensive molecular fingerprints but does not inherently guarantee source attribution without additional assumptions. Please clarify why this approach is particularly suited for identifying such relationships.

4. Figure 1: The distinction between aromatic and aliphatic compounds is shown in Figure 1, but the manuscript does not clearly explain how this classification is derived from Xc values. Please describe the Xc thresholds.

5. Lines 217-219: "Retrieving light-absorption information via a molecular-derived proxy, we use the DBE/#C ratio to identify compounds of light-absorbing aerosols (LAAs), setting limits for this range between 0.5 (for polyenes) and 0.9 (for fullerene-like hydrocarbons) (e.g., Lin et al., 2018; Laskin et al., 2015)."
As mentioned in Lin et al., 2018, the DBE/#C proxy has mainly been validated for APPI measurements, where most detected compounds fall within the BrC domain, whereas ESI positive mode typically detects a much smaller fraction of compounds in this region. Since this study relies on ESI, please discuss the potential bias introduced by ionization selectivity and how undetected BrC compounds may affect the interpretation of light-absorbing aerosol contributions.
Please consider whether complementary MS/MS structural information could support the BrC interpretation, for example by identifying characteristic functional groups or neutral losses (e.g., aromatic nitro or nitroso groups, heteroaromatic N-containing rings, conjugated carbonyl systems, or diagnostic losses such as $NO_2$, $HNO_2$, $CO$, or $CO_2$) that have been previously associated with light-absorbing chromophores.

6. Lines 228-232: "When averaging intensity by compound families (Figure S2), it is observed that BrC at the urban site contributes more strongly to nTSI, mainly due to CHN species (8.8 % vs 3.7 % at MI and SKI, respectively, in summer; 17.1 % vs 7.9 % in winter), whereas the agricultural site is more influenced by CHNO (23.9 % vs 33.9 % and 46.2 % vs 51.5 % in summer and winter, respectively)."
It is unclear why the discussion focuses exclusively on CHN species when interpreting BrC contributions, while other compound families (e.g., CHNO or CHOS) are not discussed. Please clarify whether CHN compounds dominate BrC by definition or by observation in this dataset.

7. Lines 296-298: "According to the temporal patterns, three clusters at both sites showed a clear increase in intensity during the colder season. This could suggest either a source activation during the winter, an amplification of the impact due to a lowering of the mixing layer height, a gas-to-particle partitioning phenomenon, or photochemical degradation during summer."
The logic here is unclear, as the first sentence refers to colder-season enhancement while the explanation invokes summertime photochemical degradation. Please clarify whether the authors propose enhanced wintertime sources/accumulation or active summertime photochemistry removing these compounds or their precursors.

8. Lines 335-338: "Many tracers attributed to traffic exhaust in the literature are found here: e.g. $C_4H6O_4$ (succinic acid, L4, Lui et al. (2023)), $C_7H_{12}O_7$ (L4, Thoma et al. (2022)), and $C_9H_8O_4$ (methylphthalic acid, L4, Ikemori et al. (2021)). In addition, N-containing features enrich the so-called "MI-TrOA" (Traffic OA) cluster, including aliphatic amines such as $C_8H_{18}N$, $C_7H_{17}N$, $C_9H_{21}N$ (as reported in Cao et al. (2023))."
The term "TrOA" is introduced here for the first time, although it already appears in earlier figures (e.g., Figure 3 and Figure S7). Please introduce and define this terminology consistently at its first occurrence to avoid confusion.

9. Figure S10 is not cited in the main text. Please check that all supplementary figures are properly referenced and that the numbering and order of Figures S9-S18 are consistent between the manuscript and the SI.

10. Lines 411-414: "Many of these were also listed in Zhang et al. (2023a), and attributed to wood burning (L4). Although the identification level is only 4 (unequivocal molecular formula), we hypothesize that these compounds are rising during summertime from oxidized BBOA, because of the meteorological and climatic conditions, such as the higher concentrations of oxidants species ($O_3$, $NO_3$ and OH radicals) induced by higher solar activity in this season."
This interpretation appears contradictory and requires clarification. Moreover, attributing increased $NO_3$ radical concentrations to higher solar activity is chemically incorrect, as $NO_3$ is rapidly photolyzed during daytime, and this explanation should be revised.

11. Lines 428-430: "Due to the large number of undetected compounds in the SKI-BSOA and in other clusters, the temporal pattern and the mix of both biogenic and anthropogenic SOA, we can only speculate that this cluster has a contribution due to VCP SOAs."
This attribution appears highly speculative, as excluding biogenic and anthropogenic SOA does not necessarily imply a VCP origin. Please provide stronger evidence or rephrase this statement to better reflect the uncertainty.

12. Lines 451-455: "Many of the molecular formulae the authors have found in their chamber experiments were found at both MI and SKI sites, mainly attributed to BSOA clusters, likely due to the increasing availability of $H_2SO_4$ during summertime. Nevertheless, most of the S-containing com- pounds attributed to SKI-Agricultural activities are not detected at MI site. Thus, we speculate that isomers with these formula, such as $C_4H_8O_8S$, $C_8H_{16}O_8S$, $C_{16}H_{26}O_3S$, $C_7H_{12}O_7S$ (L4), could be attributed to agricultural tractors emissions."
The attribution here is unclear and potentially contradictory. Please clarify whether identical molecular formulas appear in multiple clusters or whether different isomers with the same formulas are classified differently and provide justification or references for attributing these compounds to agricultural tractor emissions.

13. Lines 461-463: "Moreover, observing the relation between the chromatographic retention time and the molecular mass, we infer that a subset of these features is likely due to dimers from high-$NO_x$ terpene oxidation Thoma et al. (2025)."
Please check and standardize the reference formatting throughout the manuscript.

14. Lines 508-510: "Figure 5 illustrates the contribution to the monthly nTSI of each cluster at both sites. As expected, during wintertime the nTSI explained by the site-specific compounds (in darker colors) is lower than in the other seasons."
Please clearly indicate in the figure legend which colors correspond to site-specific compounds, as this is not currently evident.